# "Actually, even me I wouldn't think that it is there" exploring the knowledge and attitudes of health professionals towards autism spectrum disorders in Uganda

**Rosco Kasujja**[1]*, **Barbara Yeko Kissa**[1], **Tara Murphy**[2], **Kirstie Fleetwood-Meade**[3]

**1** Department of Mental Health and Community Psychology, Makerere University, Kampala, Uganda,
**2** Institute of Child Health, Great Ormond Street Hospital For Children NHS Foundation Trust, Tourette Syndrome Clinic, University College London, London, United Kingdom, **3** Avon and Wiltshire Mental Health Partnership NHS Trust, Child and Adolescent Mental Health Services, Bath, United Kingdom

* rosco.kasujja@mak.ac.ug, rosokasug@gmail.com

## Abstract

Globally, the prevalence of ASD is reported to be increasing requiring more health professionals to provide care. However, to-date, there is little research from Uganda. This study aimed to explore health professionals' knowledge and attitudes in the identification, diagnosis, and management of ASD in children and adolescents in Uganda. A semi-structured interview guide was used to collect data from 20 health professionals working with children and adolescents in three regional hospitals in Uganda. The data was analyzed using thematic analysis. Perceptions and understanding by health professionals about ASD were captured as three themes: competence of health professionals in identifying and managing ASD; constraints to the identification and management of ASD; and, optimism. Findings suggest there are challenges in the identification and management of ASD, which results in the low detection of ASD in Uganda. Delivery of education to the wider community and health professional training is recommended.

**Data Availability Statement:** The data has been provided within the manuscript.

## Key practitioner message

Detection and identification of ASD in children and young people is limited in Uganda.

Health professionals in Uganda experience challenges that impact on their ability to appropriately diagnose and support individuals with ASD.

Lack of knowledge has been cited as a factor in the caregivers' health seeking behavior.

The low numbers of caregivers seeking help for the children with ASD from the health centers implies that some community members still have faith in the medical field.

The sensitization of both professionals and community about the treatment and available services was recommended as a powerful tool in improving health-seeking behavior

**Funding:** The authors did not receive any funding for publication of this article

**Competing interests:** The authors have declared that they have no competing or potential conflicts of interest

## Introduction

Autism spectrum disorder (ASD) is a group of neurodevelopmental disorders characterized by deficits in social communication and interaction and the presence of restricted and repetitive patterns of behaviors, interests, and activities with onset in the first three years of life [1]. The cause of ASD is thought to be a combination of genetics and environmental factors [2]. Global, ASD is reported to be increasing [3–5]. To date, prevalence rates of ASD in Africa and other low and middle-income countries (LMIC) is unclear [6–8]. However, a few comparative studies have demonstrated that the prevalence of common mental health disorders and intellectual disability show similarities across the globe [9].

Uganda has one of the youngest populations in the world with 31% of the 34 million population aged between 10–24 years [10]. Estimates suggest that up to 35% of all Ugandans suffer from a mental disorder and 15% of the population require treatment [11]. The prevalence rate of ASD in Uganda is estimated to be 1.2–1.3/100 among children aged 2–9 years [12]. Mental health services in Uganda have been described as inadequate [11] because of limited funding which stands at less than one percent of the national budget [13].

As is in most African countries, mental health services in Uganda are mainly situated around the city leaving the rest of the rural population with limited or no access [14]. Additionally, a lack of specialized health personnel and specialist care facilities have resulted in challenges in the diagnosis of ASD [15–17].

Considering the emerging understanding of neurodevelopmental needs of children and young people in Uganda, there's a timely opportunity to understand the factors that impact identification, diagnosis and management of ASD. The findings will contribute to novel knowledge and understanding to guide service development. Therefore, this study sought to explore health professionals' current knowledge and attitudes towards ASD in children and adolescents in Uganda.

## Materials and methods

The study was conducted across three regional referral hospitals in Eastern Uganda, namely Soroti, Mbale, and Jinja, where children and adolescents with Autism Spectrum Disorder (ASD) utilize various hospital services. All health professionals working directly with this population who provided consent were included in the study, resulting in a sample size of 20 participants, as detailed in Table 1. Recruitment of respondents occurred between February 5th, 2019, and February 15th, 2019.

### Procedure

A semi-structured interview guide with open-ended questions was administered to the respondents. The interview aimed at eliciting professional knowledge regarding the diagnosis, treatment, and management of ASD following the DSM 5 diagnostic criteria for ASD.

### Ethical considerations

This study adheres to rigorous ethical standards and guidelines set forth by Makerere University School of Health Sciences Research and Ethics Committee, Kampala, Uganda (SHSREC REF NO: 2018–042), which provided ethical approval for the research endeavour. The participants involved in this study provided formal written consent before their involvement, ensuring their voluntary participation and understanding of the study's objectives, procedures, and potential risks. The consent process was conducted under the principles outlined by the SHSREC 2018–042 IRB reference number, prioritizing the autonomy and welfare of the participants.

**Table 1. Participants' background information.**

|  | Profession | Age | Sex | Qualification | Years of clinical practice |
|---|---|---|---|---|---|
| 1 | Psychiatric Officer | 50 years | Male | Diploma | 15 years |
| 2 | Psychiatric Officer | 34 years | Female | Diploma | 10 years |
| 3 | Occupational Therapist | 26 years | Male | Diploma | 4 years |
| 4 | Nurse | 46 years | Female | Degree | 20 years |
| 5 | Psychiatric Officer (CAMHS) | 33 years | Male | Diploma | 4 years |
| 6 | Enrolled Nurse | 29 years | Female | Diploma | 6 months |
| 7 | Enrolled Nurse | 25years | Female | Certificate | 1 month |
| 8 | Nutritionist | 47 years | Male | Degree | 19 years |
| 9 | Psychiatric Clinical Officer | 49 years | Male | A Level | 20 years |
| 10 | Psychiatric Clinical officer | 51 years | male | Degree | 23 years |
| 11 | Psychiatric clinical officer | 45 years | male | Degree | 14 years |
| 12 | Pediatrician | 40 years | Female | Masters | 14 years |
| 13 | Nutritionist | 51 years | Male | Masters | 14 years |
| 14 | Senior Nursing Officer | 50 years | Female | Diploma | 1 month |
| 15 | Nurse | 52 years | Female | Degree | 9 years |
| 16 | Pediatrician | 33 years | Male | Masters | 6 years |
| 17 | Clinical Officer | 25 years | Male | Diploma | 2 months |
| 18 | Nurse | 56 years | Female | Diploma | 20 years |
| 19 | Medical Officer | 29 years | Male | Degree | 3 years |
| 20 | Psychiatric Clinical fficer | 44 years | Female | Degree | 5 years |

## Analysis

Transcripts from the audio recordings were transcribed verbatim and anonymized. The interviews were analysed using thematic analysis. Two authors (RK, BYK) read through transcripts several times, with each independently noting down any connections, summaries, and or preliminary interpretations. The same was done for coding. Connections between themes were cross-referenced back to the original transcripts, to ensure that participants' original accounts had not been lost or misrepresented through the process of interpretation. Further cross-checking was utilized during the writing process to remain close to the data and finally a coherent narrative of the analysis was written [18].

## Results

Data is presented through themes and sub-themes.

## Challenges with detection

This theme highlights challenges that health professionals reported in detecting ASD. Two sub-themes emerged:

### Inadequate expertise to diagnose

Most health professionals that all clinics they worked in assessed children for ASD. However, there were varying rates of expertise of ASD. Whilst some health professionals described ASD as a neurodevelopmental disorder or a brain disorder; others described it as a behavioral disorder. One participant (Mary) said:

> "*You know Autism, these are . . . behavioral disorders . . . just train them. . . . . .*"

Francis added that:

*"..You find that the client has been having some odd behaviors for some time but as a result of the behaviors, we now maybe manage substance or oppositional behaviour. . .In my practice I think I have missed out Autism"*

Others described ASD symptoms as childhood schizophrenia, whilst other participants described hallucinations and bipolar mood swings as key symptoms to look out for when identifying ASD.

Masitula explained:

*"I think there are confusing opinions. . .what I know is that it's just schizophrenia in children"*

Two participants described poor diet and lack of folic acid supplements during pregnancy as a cause of ASD. Hypoxia at birth was also looked at as a cause of ASD.

Agnes explained:

*"..Sometimes they feed on more of carbs than the others [food groups]. They even get kwashiorkor (severe malnutrition manifested through physical symptoms) . . . When they get those problems, they also they end up getting . . .. Autism also"*

Only four participants described the symptoms of ASD as occurring along a spectrum. This may be because professionals were more likely to see children with significant ASD symptoms clinically, whilst children presenting with mild levels of ASD are not brought for care due to

## Constraints on the diagnosis of ASD

Health professionals face constraints while working with children with ASD. These constraints are presented in the subthemes below:

### Lack of exposure

A majority of the participants regardless of the extent of their experiences, indicated that ASD was a condition they rarely encountered. On their entire career they reported seeing one or two patient(s).

Mary said:

*"I wouldn't think that it [ASD] is there, only that I have studied about it, at least I know something small about it"*

Stella added:

*"So far since this year began, we saw only one case of ASD they are few, they are rare cases"*

Therefore, the inadequate skills to recognise mild ASD may underlie the under-detection. At the same time, the belief that is ASD is rare may also be the cause for low levels of diagnosing.

### Lack of specialized training in ASD

The majority of the health workers reported that they had limited training on ASD as it was regarded as a rare illness.

David explained:

*". . .because when you are in school. . .. we did not even study Autism, we just by-passed it. . .one day our lecturer talked about it. . .but he was like 'you might not find it in all your years of practice"*

Whilst Tito added:

*"Inadequate training on ASD during medical school and presumably no further opportunities to train once I qualified"*

Standards

If at the training stage, professionals are not fully taught how to identify ASD symptoms accurately and how often to look for it, then this presents a significant barrier.

## Time constraints

Professionals disclosed that making a diagnosis of ASD was very difficult and confusing because it took more than one or two sessions to diagnose. They also reasoned that behavior and symptoms needed to be observed over time.

Stella expressed this difficulty:

*"Yes, it is difficult. . . because they are rare, you take long coming up with that diagnosis, it's a challenge at times. First you might misdiagnose with other childhood disorders and when they come back that's when the second time on review that's when you can know that it is not"*

In a setting where a health professional has many patients waiting to be seen in clinic, making time for an individual who needs additional time for assessment is a constraint.

## Limitations in managing ASD

Participants highlighted many roadblocks which made it more challenging for practitioners to both create and implement management plans with families to support the young people with ASD. These are explored below within the subthemes of: Financial difficulties, cultural perceptions and stigma, and adverse previous hospital experiences.

## Financial difficulties

Health workers noted that the treatment of ASD requires multiple visits to the hospital and a treatment plan. The ongoing care demands deter parents from seeking support for the treatment of ASD because the majority of the population are poor and cannot afford to pay for travel (which may be lengthy distances).

Faith said:

*". . .these parents are broke. You are sending them, buy Ritalin, buy toys, is that in their budget really? They do not have money for food, and you are saying buy toys? So, those are the challenges. So, for you, you may be having high expectations, but the expectations are not really what the parent can afford"*

Furthermore, it was noted that families were often unable or unwilling to return to clinics for multiple appointments, which made the implementation of behavioural interventions very challenging for health workers.

Tito said:

"... most of my management of course is psychological treatment ...and behavioral treatments which take a lot of time, yet caregivers were unable or not willing to spare time for this"

## Cultural perceptions and stigma

Health workers noted that in the local communities, ASD was not viewed as a medical issue but as a curse, being bewitched, or possessed by evil spirits. Management of ASD required attendance church, mosque, or visiting a traditional healer. Attendance at hospital was often the last option, and families may have been on a long journey to understand their child's symptoms and experiences.

Tito explained:

*"People first start with church, they believe so much in evil spirits, or in God I don't know... If church fails, they go to witch doctors, if it fails, they end up in the hospital"*

Some participants highlighted how certain families may be stigmatised and avoided due to beliefs Henry said:

*"Up to now, people still point fingers; that home if you go there to marry, you will produce those [give birth to a child with ASD]...so that's why when they are there, they want to keep them such that no other person knows, even me, it took me some time to know [that my neighbor had a child with ASD] yet we live just opposite..."*

## Adverse hospital experiences

Past negative experiences at the hospital were cited by the health professionals as one of the reasons some caregivers refused to go to the hospital. For example, long queues, not receiving any help during a visit, and being asked to pay (bribes) for medical services in a government hospital, which should offer free services. These experiences disrupt health seeking behavior which in turn disrupts clinical care. Participants reported that families then resort to seeking help from other non-medical alternatives or resort to smaller clinics, where they are likely to receive help from less specialist practitioners.

David said:

*"..there are some medical personnel or health professionals who always ask money from these people and so sometimes you find they don't even want to come...Remember these are places where there are good doctors who can identify these problems and they go to clinics where people are looking for even more money"*

David highlights how families in Uganda may not be aware of the best place to go to receive a diagnosis (such as larger clinics, where there are more specialist staff).

## Optimism

It was clear from many professionals that there was hope and a strong belief that children with ASD (and their families) could experience an improvement to their situation and the management of their symptoms. This is explained within the following 3 subthemes.

## Hope for improvement

All participants indicated that children with ASD could experience an improvement in symptoms, especially with the use of behavioural therapy. Participants felt behavioural therapy would enable children with ASD to live a more independent life, through making treatment interactive and breaking down treatment goals into small workable units that encourage progress. Mary said:

*"If they are not talking you just involve them in the talking just continue doing it, eventually they will catch up".*

This implies that despite the length of time it takes to detect ASD and the difficulties in managing ASD, the health workers were optimistic that with persistence they can offer a meaningful service to families who seek treatment.

## Desire to learn more

Professionals expressed a need specifically for more knowledge on management guidelines, novel treatments, for example, the use of stem cell therapy, and how to manage treatment of children better within the community. Other participants wanted to know more about the definition and nature of ASD because they felt confused about it. The quote below from simon exemplifies the professionals' desire for new knowledge:

*". . . . ...for autism, I think the spectrum definition is wide and we need to know much more and when we get knowledgeable about it, we can see which care to give, like for example we can try to know their nutritional needs and integrate nutrition into their care"*

## Belief in the power of community sensitization

All participants were positive that sensitisation/awareness activities would improve awareness amongst health workers and the communities. Psychoeducation is an effective tool in increasing knowledge and correcting misperceptions about ASD. Support supervision and incorporating mental health into training programs and medical school curriculum will encourage the diagnosis and management of ASD.
Henry said:

*". . .majority of them [children with ASD] are in the community, they have not reported here [to medical settings]. So, if we moved into the community with community leaders, and we identify them in the community and we manage them in the community I think we would have done something great"*

The results clearly highlight the constraints that the children and families affected by ASD grapple with, while at the sometime underlining the need of adequate skills in the diagnosis, treatment and management of ASD and other commodities.

## Discussion

The aim of the study was to explore the knowledge and attitudes of a wide range of health professionals of ASD in Uganda. Results indicated that in everyday clinical practice, health professionals reported inadequate knowledge as a challenge in identification and management of ASD. This is consistent with findings from Nigeria, other Sub-Saharan countries, and China

[19–22] who found that medical staff reported lower knowledge about ASD as compared to other healthcare professionals. In Bakare's study, the knowledge gap was attributed to significant inadequacy in medical school curriculum in preparing professionals to manage ASD whereas in Uganda it was attributed to limited time allocated to teaching ASD while in school. Similar reports come from high income countries such as the findings of [23] in Canada, and [24,25] in the USA.

Perceived limited exposure to ASD cases in the current study was reported as a barrier to developing expertise in this area. This finding agrees with the findings of [26,27] who found that health professionals reported ASD as rare and therefore difficult to manage. However, it is likely that this is tautological in that perception of limited exposure to children with ASD results from the fact that health professionals are not sufficiently trained to identify cases. This highlights the need for funded epidemiological research in both the community and clinical services in Uganda.

Due to inadequate knowledge, professionals in the current study held varying perceptions about the etiology of ASD; some professionals considered poor nutrition, birth injuries and failure to attend antenatal care services as the main cause of ASD in children. This is consistent with the findings of [28,29] who concluded that most professionals believed that the etiology of autism can be natural, supernatural, and preternatural. From a Western medicine viewpoint, such beliefs may result in conclusions from history taking as leading to a misdiagnosis or confusion on the part of the practitioner.

Professionals also recognized that due to inadequate skills and knowledge on the diagnosis and management of ASD, they had misdiagnosed ASD and recommended inappropriate treatment. This is supported by the findings of [30,31] who concluded that the lack of knowledge of ASD among staff can lead to misdiagnosis, delayed diagnosis, and delayed intervention. The findings of the study may offer an explanation to the poor health-seeking behavior of caregivers who seek alternative treatment options [20].

The findings of this study indicate that most health professionals used difficulty in social communication and social interaction as key diagnostic criterion for ASD. This suggests that a comprehensive training of the few health professionals is much needed so that diagnosis is early and accurate.

The findings agree with studies by [32,33] who concluded that health professional possessed inadequate knowledge to properly diagnose and manage ASD. However, the focus of health professionals on severe destructive symptoms could offer a tentative explanation for lower detection rates in calm children as compared to children who exhibited aggressive behavior. Further, misdiagnosis could be an explanation as to why some parents do not return to the hospital and seek care elsewhere.

Participants reported that caregivers and the community held a belief in the spiritual cause and treatment of ASD. This was reported to negatively impact health-seeking behavior and compliance to treatment because caregivers considered medical help as a last resort and this in return made the treatment and management process of children with ASD very difficult because of poor consistence health seeking patterns. This pattern is widely established in the research literature [8,34–37]. This dominant belief in the spirituality of ASD could explain the community stigma [38,39] towards people with ASD and poor health-seeking behavior. For example, children with ASD in Ethiopia were sometimes chained up at home [40].

Presence of impairing co-occurring conditions plays a major role in seeking medical help. The health professionals argued that most caregivers brought children to the hospital for conditions other than ASD and that ASD was recognized as they treated other illnesses. This trend is consistent with the findings of [34,41] who found that most parents initially took their children to family doctors and pediatricians because of concern over comorbid conditions.

However, comorbid conditions made it difficult to manage ASD. This is consistently evident in existing literature (see [42–45]) indicating that children with ASD present with an array of mental and non-mental conditions that make the detection and management of ASD complex.

Participants were hopeful in discussing how engaging families in meaningful work (such as behavioural interventions) was possible. Over time, a positive ripple effect through communities witnessing these changes may even combat the negative perception of and stigma towards ASD that families and the community have. Furthermore, comprehensive training for mental health professionals in managing ASD is imperative. Future studies will benefit from focusing on the knowledge, experiences and attitudes of spiritual leaders, parents/caregivers and of educators in both mainstream and special needs schools to get a holistic picture of the child's context.

A future country-wide Ugandan study would give a more representative picture. The sample was small and therefore results from this study cannot be inferred to describe the general population of health professionals in Uganda.

## Educational implications

Education institutions have a significant influence on how future health professionals think and behave. In order to bridge the knowledge gap revealed by this study, educational establishments had to give top priority to including thorough curriculum on childhood diseases, with an emphasis on autism spectrum disorder in particular (ASD). Competency-based education would be beneficial for health institutions since it has been found to significantly increase practitioners' knowledge[46,47]. This entails providing a wide range of healthcare workers, such as nurses, doctors, and physiotherapists, with comprehensive understanding of ASD, including diagnosis, referral procedures, and evidence-based treatments adapted to the Ugandan environment. Educational institutions should also encourage cooperative partnerships with families who are raising children with ASD. Educational institutions may eliminate stereotypes and prejudices about ASD in Ugandan communities by exposing students to these families directly. This proactive approach ensures that future educators and practitioners are well-informed and culturally sensitive, contributing to a healthcare workforce that is both knowledgeable and empathetic towards the unique needs of individuals with ASD.

The report also emphasizes how important it is for health institutions and the communities they serve to work together more closely. There is a serious communication gap between the public and healthcare practitioners, as evidenced by the stigma and misunderstandings about ASD that have been seen in Ugandan communities. Educational establishments can effectively contribute to closing this divide by supporting community involvement programs. This might be mutual visits, whereby people of the community visit medical institutions and medical experts provide instructional programs in nearby villages. These seminars ought to concentrate on developmental impairments, offering correct ASD knowledge and sharing top management techniques. These community-hospital collaborations have the power to change perceptions, lessen stigma, and encourage proactive health-seeking behavior. By fostering mutual learning between healthcare providers and community members, educational institutions can contribute to a more informed and inclusive approach to the care of individuals with ASD and other developmental disorders in Uganda.

## Conclusions

The study revealed a pattern of professionals having challenges that impact on their ability to appropriately diagnose and support individuals with ASD. The lack of knowledge in turn

might negatively impact on the caregivers' health seeking behavior because of not getting the help they are seeking. The low numbers of caregivers seeking help for the children with ASD from the health centers implies that some community members still have faith in the medical field. However, sensitization of both professionals and community was cited as a powerful tool in breaking this pattern.

## Author Contributions

**Conceptualization:** Rosco Kasujja, Barbara Yeko Kissa, Tara Murphy, Kirstie Fleetwood-Meade.

**Data curation:** Rosco Kasujja, Tara Murphy, Kirstie Fleetwood-Meade.

**Formal analysis:** Rosco Kasujja, Barbara Yeko Kissa, Tara Murphy.

**Investigation:** Rosco Kasujja.

**Methodology:** Rosco Kasujja, Barbara Yeko Kissa.

**Project administration:** Rosco Kasujja.

**Resources:** Rosco Kasujja, Tara Murphy.

**Supervision:** Rosco Kasujja, Tara Murphy, Kirstie Fleetwood-Meade.

**Validation:** Rosco Kasujja, Tara Murphy.

**Writing – original draft:** Rosco Kasujja, Kirstie Fleetwood-Meade.

**Writing – review & editing:** Rosco Kasujja, Kirstie Fleetwood-Meade.

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
