## [Decision Letter · Decision Letter 0]

9 Apr 2024

PMEN-D-24-00075

Actually, even me I wouldn’t think that it is there” Exploring the knowledge and attitudes of health professionals towards Autism Spectrum Disorders in Uganda.

PLOS Mental Health

Dear Dr. KASUJJA,

Thank you for submitting your manuscript to PLOS Mental Health. After careful consideration, we feel that it has merit but does not fully meet PLOS Mental Health’s publication criteria as it currently stands. Therefore, we invite you to submit a revised version of the manuscript that addresses the points raised during the review process.

We look forward to receiving your revised manuscript.

Kind regards,

Juan Felipe Cardona, Ph.D.

Academic Editor

PLOS Mental Health

Journal Requirements:

1. Please amend your online Financial Disclosure statement. If you did not receive any funding for this study, please simply state: “The authors received no specific funding for this work.”

2. Please update your online Competing Interests statement. If you have no competing interests to declare, please state: “The authors have declared that no competing interests exist.”

3. In the online submission form, you indicated that "The data will be made available upon request".

a) In a public repository, 

b) Within the manuscript itself, or 

c) Uploaded as supplementary information.

Additional Editor Comments (if provided):

Dear Authors,

Thank you for submitting your manuscript for consideration. After a detailed review, it has been determined that major revisions are necessary to meet the publication standards. The reviewers have provided specific feedback, which we believe will significantly improve your manuscript's contribution and clarity.

Please review the attached detailed comments and prepare a revised version of your manuscript. Your revised submission should also include a point-by-point response to the reviewers' comments.

Reviewers' comments:

Reviewer's Responses to Questions

**Comments to the Author**

1. Does this manuscript meet PLOS Mental Health’s publication criteria? Is the manuscript technically sound, and do the data support the conclusions? The manuscript must describe methodologically and ethically rigorous research with conclusions that are appropriately drawn based on the data presented.

Reviewer #1: Yes

Reviewer #2: Partly

2. Has the statistical analysis been performed appropriately and rigorously?

Reviewer #1: No

Reviewer #2: Yes

3. Have the authors made all data underlying the findings in their manuscript fully available (please refer to the Data Availability Statement at the start of the manuscript PDF file)?

Reviewer #1: No

Reviewer #2: Yes

4. Is the manuscript presented in an intelligible fashion and written in standard English?

Reviewer #1: Yes

Reviewer #2: Yes

5. Review Comments to the Author

Reviewer #1: 1. What are the main claims of the article and what is their significance for the discipline?

The manuscript's central claim is the notable gap in knowledge and appropriate attitudes of health professionals in Uganda toward Autism Spectrum Disorders (ASD). This claim is pivotal for the discipline as it identifies critical shortcomings in healthcare services for ASD, emphasizing the need for enhanced professional training and awareness, especially in low-resource settings like Uganda. The study's findings have profound implications for healthcare policy and practices, indicating an urgent need for tailored educational and training programs in such regions.

2. Are the claims appropriately placed in the context of previous literature? Have the authors treated the literature fairly?

The study's claims are strategically placed within the ambit of previous literature, reflecting a comprehensive understanding of global challenges associated with ASD and the unique context of Uganda. The authors exhibit a balanced and fair treatment of the literature, adeptly integrating their research within the existing framework and identifying gaps that their study addresses.

3. Do the data and analysis fully support the claims? If not, what additional evidence is needed?

The qualitative data derived from semi-structured interviews substantiates the study’s claims. However, these claims could be further strengthened.

Aspects Supporting the Claims: The insights from health professionals in Uganda provide concrete evidence regarding the existing knowledge and attitude gaps toward ASD.

Aspects Requiring Additional Evidence: Additional quantitative data, perhaps in the form of a larger-scale survey, could provide further validation and a more nuanced understanding of these issues across a broader spectrum of healthcare settings.

4. If a protocol is provided, for example for a randomized controlled trial, are there significant deviations from it? If so, have the authors adequately explained why the deviations occurred?

The manuscript does not describe a randomized controlled trial protocol; instead, it utilizes a qualitative research approach. The selected methodology is appropriate for the study’s objectives, aiming to deeply explore attitudes and knowledge regarding ASD among health professionals in Uganda.

5. PLOS Mental Health encourages authors to publish detailed protocols and algorithms as online supporting information. Does any particular method used in the manuscript warrant such treatment?

Methods Justifying Detailed Publication: The utilization of semi-structured interviews and thematic analysis justifies the publication of detailed protocols. This detailed documentation could significantly aid replication efforts and offer valuable insights for similar research endeavors in other low-resource settings, thereby contributing to the global body of knowledge on ASD healthcare challenges.

6. If the article is deemed unsuitable for publication in its current form, does the study itself show enough potential to encourage the authors to resubmit a revised version?

Strengths of the Study: The study’s innovative focus on the perception of ASD in the healthcare system of Uganda and its comprehensive qualitative analysis are its core strengths.

Areas for Improvement: Enhancements such as broadening the sample size, integrating quantitative research methods, and possibly diversifying the range of healthcare settings would significantly improve the manuscript. These modifications could deepen the study's impact and merit consideration for resubmission and publication.

7. Are the original data deposited in appropriate repositories and are accession/version numbers provided for genes, proteins, mutants, diseases, etc.?

Given the qualitative nature of this research, the deposition of traditional scientific data like genes or proteins is not applicable. However, ensuring secure and accessible storage of interview transcripts and thematic analysis data, while maintaining ethical considerations for confidentiality, is crucial for the study's transparency and replicability.

8. Does the study conform to any relevant guidelines such as CONSORT, MIAME, QUORUM, STROBE, and the Fort Lauderdale Agreement?

The manuscript does not explicitly state its adherence to these specific guidelines. Nonetheless, reviewing and aligning with guidelines like STROBE, which is pertinent for observational studies, would bolster the manuscript's credibility and reflect adherence to the highest standards of research reporting.

9. Are the methodological details sufficient to allow for the replication of the experiments?

The methodological details, while adequately outlined for a qualitative study, would benefit from further elaboration on aspects such as interview procedures, participant selection, and thematic analysis. This additional detail would enhance the study’s replicability and robustness.

10.Is there any software created by the authors available for free?

The manuscript does not indicate the development or use of any specific software tools. Future iterations of the study might consider incorporating software tools for data analysis or virtual interviewing, enhancing the study's methodological rigor.

11. Is the manuscript well organized and clearly written to be accessible to non-specialists?

The manuscript is effectively organized and articulately composed, ensuring accessibility to a broad audience. Expanding upon the implications of the findings for healthcare policy and practice, and including a more detailed discussion on practical applications, would further enhance its relevance and utility to both specialists and non-specialists.

Reviewer #2: Generally, the manuscript is well written, it is clear and easy to read through. However, the following can be done to improve it;

1. The title is very long and difficult to comprehend the first section. This part can be deleted and simply state "Exploring the knowledge and attitudes of health professionals towards Autism Spectrum Disorder in Uganda"

2. The study aimed to explore knowledge and attitudes, why did the authors leave out practices as is commonly done in KAP (Knowledge, Attitude and Practices) studies?

3. How was the sample size of 20 participants determined? Was it the point of saturation?

4. Page 10 sentence 3 is incomplete "...ASD are not brought for care due to___________"

5. The section for "Education implication" is not clear. The authors need to check the tense used at the beginning of this section. It implies that the study involved educationists.

6. In the conclusion, the authors could suggest future direction of research.

6. PLOS authors have the option to publish the peer review history of their article (what does this mean?). If published, this will include your full peer review and any attached files.

**Do you want your identity to be public for this peer review?** For information about this choice, including consent withdrawal, please see our Privacy Policy.

Reviewer #1: No

Reviewer #2: No

---

## [Decision Letter · Decision Letter 1]

10 Jul 2024

Exploring the Knowledge and Attitudes of Health Professionals towards Autism Spectrum Disorder in Uganda.

PMEN-D-24-00075R1

Dear Senior Lecturer KASUJJA,

We are pleased to inform you that your manuscript 'Exploring the Knowledge and Attitudes of Health Professionals towards Autism Spectrum Disorder in Uganda.' has been provisionally accepted for publication in PLOS Mental Health.

Best regards,

Juan Felipe Cardona, Ph.D.

Academic Editor

PLOS Mental Health

Dear Author

I wanted to inform you that I have reviewed your responses to the reviewer comments for the manuscript. Your revisions have been thoroughly addressed, and I believe the manuscript now meets the necessary criteria for acceptance.

Thank you for your diligent work and cooperation throughout this process.

Best regards,

Juan Felipe Cardona

Reviewer Comments (if any, and for reference):

Reviewer's Responses to Questions

**Comments to the Author**

1. If the authors have adequately addressed your comments raised in a previous round of review and you feel that this manuscript is now acceptable for publication, you may indicate that here to bypass the “Comments to the Author” section, enter your conflict of interest statement in the “Confidential to Editor” section, and submit your "Accept" recommendation.

Reviewer #2: All comments have been addressed

2. Does this manuscript meet PLOS Mental Health’s publication criteria? Is the manuscript technically sound, and do the data support the conclusions? The manuscript must describe methodologically and ethically rigorous research with conclusions that are appropriately drawn based on the data presented.

Reviewer #2: Yes

3. Has the statistical analysis been performed appropriately and rigorously?

Reviewer #2: Yes

4. Have the authors made all data underlying the findings in their manuscript fully available (please refer to the Data Availability Statement at the start of the manuscript PDF file)?

Reviewer #2: Yes

5. Is the manuscript presented in an intelligible fashion and written in standard English?

Reviewer #2: Yes

6. Review Comments to the Author

Reviewer #2: The manuscript has been improved, the authors have addressed my comments.

My last comment before publication is for the authors to ensure consistency of the diagnosis as per DSM 5 i.e they should stick to "Autism Spectrum Disorder" and avoid using the diagnosis "Autism" which was for DSM IV.

7. PLOS authors have the option to publish the peer review history of their article (what does this mean?). If published, this will include your full peer review and any attached files.

**Do you want your identity to be public for this peer review?** For information about this choice, including consent withdrawal, please see our Privacy Policy.

Reviewer #2: No
